# The Fast and the Furriest: Investigating the Rate of Selection on Mammalian Toxins

**DOI:** 10.3390/toxins14120842

**Published:** 2022-12-01

**Authors:** Leah Lucy Joscelyne Fitzpatrick, Vincent Nijman, Rodrigo Ligabue-Braun, K. Anne-Isola Nekaris

**Affiliations:** 1Nocturnal Primate Research Group, Department of Social Sciences, Oxford Brookes University, Oxford OX3 0BP, UK; 2Centre for Functional Genomics, Department of Health and Life Sciences, Oxford Brookes University, Oxford OX3 0BP, UK; 3Department of Pharmacosciences, Federal University of Health Sciences of Porto Alegre (UFCSPA), Avenida Sarmento Leite 245, Porto Alegre 90050-130, Brazil

**Keywords:** mammals, selection rates, dN/dS, venom evolution, Primates, Eulipotyphla, Chiroptera, Monotremata

## Abstract

The evolution of venom and the selection pressures that act on toxins have been increasingly researched within toxinology in the last two decades, in part due to the exceptionally high rates of diversifying selection observed in animal toxins. In 2015, Sungar and Moran proposed the ‘two-speed’ model of toxin evolution linking evolutionary age of a group to the rates of selection acting on toxins but due to a lack of data, mammals were not included as less than 30 species of venomous mammal have been recorded, represented by elusive species which produce small amounts of venom. Due to advances in genomics and transcriptomics, the availability of toxin sequences from venomous mammals has been increasing. Using branch- and site-specific selection models, we present the rates of both episodic and pervasive selection acting upon venomous mammal toxins as a group for the first time. We identified seven toxin groups present within venomous mammals, representing Chiroptera, Eulipotyphla and Monotremata: KLK1, Plasminogen Activator, Desmallipins, PACAP, CRiSP, Kunitz Domain One and Kunitz Domain Two. All but one group (KLK1) was identified by our results to be evolving under both episodic and pervasive diversifying selection with four toxin groups having sites that were implicated in the fitness of the animal by TreeSAAP (Selection on Amino Acid Properties). Our results suggest that venomous mammal ecology, behaviour or genomic evolution are the main drivers of selection, although evolutionary age may still be a factor. Our conclusion from these results indicates that mammalian toxins are following the two-speed model of selection, evolving predominately under diversifying selection, fitting in with other younger venomous taxa like snakes and cone snails—with high amounts of accumulating mutations, leading to more novel adaptions in their toxins.

## 1. Introduction

The advent of venomous animal transcriptomes and genomes, advances in proteomics and the reduction in costs for next-generation sequence technology has allowed advances in the study of venom evolution [1,2]. Venom is an extraordinary trait to document the mechanics of evolution, having been identified within >250,000 animal species, with over 11 different functional roles and posited to have evolved independently in 101 different animal lineages [3]. In addition, the thousands of identified toxins within animal venoms and different venom delivery systems provide examples of parallel and convergent recruitment across and within clades [4].

Random mutations in coding genes that lead to variation within traits are what enables species to adapt to their environments. Novel mutations may become fixed within an organism or population if they provide an advantageous adaption against various selection pressures encountered throughout an organism’s life (i.e., predation, sexual selection or availability of resources) [5]. Measuring the rate of synonymous mutations (known as dS) in comparison to neutral mutations (known as non-synonymous; dN) found within a gene or protein of interest can allow researchers to infer if adaption is occurring and which selection pressures may be responsible. Toxins from venomous animals are particularly interesting to evolutionary biologists as they often contain a high proportion of synonymous mutations and are observed to be some of the fastest evolving genes on the planet due to their rapid accumulation of synonymous mutations over a relatively short evolutionary time [6,7,8]. In addition, toxins are often represented with multiple isomers or homologous sequences within a venom—in part due to the neo- or sub-functionalisation of toxins recruited into the venom from duplication of non-toxin genes. These homologous sequences can be well conserved between species, families or even orders [9]. It also leads to a greater observed diversity of functions and prey targets due to these homologous sequences.

In 2015, Sungar and Moran examined the rate of selection acting on toxin groups available and found a link between the evolutionary age of a venomous taxon and the rate of selection acting on their toxins [9]. Older groups (e.g., Chilopoda, Cnidaria and Araneae) had a higher percentage of purifying (dS < dN) selection on their toxins, in comparison to younger groups (e.g., Conidae and Serpentes), which had greater diversifying (dS > dN) selection acting on their toxins. The authors presented this as a ‘two-speed’ mode of evolution within toxins: that a rapid and diversifying recruitment of toxins is then followed by a retention and stabilising of toxins under purifying selection. A drawback of the study was that the data had come exclusively from predatory venoms—the study used no data from either mammals or fish.

In mammals, venom represents a unique model to investigate evolution, as it has evolved independently in eight lineages in four different clades with five different functional uses, the most diverse of any other venomous group [10,11] (Figure 1, Table 1).

These uses include intraspecific, both sexual (*Ornithorhynchus anatinus*) [31] and territorial (*Nycticebus* spp. and *Xanthonycticebus pygmaeus*) [32], haematophagy (Desmodontinae) [33] and food storage (Eulipotyphla) [13]. Currently, there are over 100 different molecules (both toxic and non-toxic) identified from mammal venoms. The exact purpose of many of these molecules to their target organisms remains unclear, due to the small amount of venom produced by many of these animals, difficulty in collecting samples from rare and endangered taxa, or difficulty in replicating the exact function of these venoms (particularly for intraspecific usage).

The venom of the platypus has been well documented in part due to the platypus being one of the first mammals to receive a full genome assembly [14,34]. The venom of the platypus is noted for being highly painful although no human deaths are recorded. Platypus venom is exclusively found within males during the breeding season (August to October, January to March) [35]. The most recognisable toxins found within platypus venom are the OvDLPs (defensins), CRiSPs (known as Cysteine-rich secretory proteins) and Kunitzs, all thought to play a role in producing pain via neurotoxicity [36].

Within Chiroptera, all molecules so far identified within the venom are assumed to help facilitate blood feeding: plasminogen activator prevents blood from clotting to allow a constant flow of blood when feeding [24]. Other toxins found within Chiroptera venom include CRiSP (which are proposed to either have a peripheral vasodilatory effect or be potentially neurotoxic), Kunitz domains (suggested to have anticoagulation properties), Kallikrein-1 (known as KLK1, a serine protease, suggested to aid in cleaving fibrogen which prevents blood clotting and vasodilation), Pituitary Adenylate Cyclase-Activating Polypeptide (known as PACAP, proposed to aid in vasodilation). In addition, *Desmodus* allergen-related lipocalins (called Desmallipins) have been identified within the venom although it is not clear what purpose they provide [23].

Within Eulipotyphla, the most prominent and well-documented venom is a Kallikrein-1 derivative (KLK1). Eulipotyphla paralyse their prey and store them in food larders. Due to the exceptionally high metabolism found within members of Eulipotyphla, they must eat every few hours [13]. KLK1 in shrews is thought to cause a drop in blood pressure and in turn result in paralysis within prey—both invertebrate and vertebrate [37].

The only protein identified within slow loris venom so far is a secretoglobin called Brachial Gland Secretion (BGE). This protein must be mixed with slow loris saliva to become toxic, being the only known two-step venom observed in the animal kingdom so far [26]. The BGE is a secretoglobin that shares a homology with the Fel-d-1 protein produced by domestic cats (*Felis catus*) that causes allergies in humans—humans bitten by slow lorises often display anaphylactic shock. Slow lorises are highly territorial, with venom used in intraspecific competition between females or between males, resulting in necrotic wounds that can lead to intraspecific injury or death [32].

Despite this convergent origin of venom in mammals, previous work has identified shared toxins and molecules found within mammal venoms including kallikrein-1 (Eulipotyphla and Chiroptera) [16,17,23], secretoglobins (Primates, Eulipotyphla and Chiroptera) [17,23,25] and CRiSPs (Platypus and Chiroptera) [23,38]. Venom evolving convergently within and across the class, as well as having multiple different functional usages but still with shared toxins is distinct in mammals amongst venomous animals. Several mammal toxins are represented by multiple paralogs or isomers, which allows an opportunity to investigate if selection has acted equally across sequences within a toxin group or if there is evidence of episodic selection promoting variation within similar sequences.

Several researchers have investigated selection of mammalian toxins, but these papers have been published intermittently over two decades, using different methodology and different criteria to test hypotheses. The availability of mammalian toxin data has also grown, allowing an opportunity to not just test species evolution but the evolution of toxins across the whole Class, affording a direct comparison. This presents the opportunity to investigate if there are patterns of selection across the Class due different factors such as the functional use of venom, the evolutionary age of the group or the type of toxin. This direct comparison between different venomous mammal groups is a novel contribution to the literature.

By using multiple representatives of mammalian toxins, we can begin to answer three questions pertaining to mammal toxin evolution:What is the dN/dS ratio of mammal toxins and where does the Class fit within the ‘two-speed’ model of evolution proposed by Sunagar and Moran (2015) [9]?Are sites found under selection subject to pervasive selection (found across the whole phylogeny) or episodic selection (found within a subset of branches)? Is there evidence that different branches (toxin sequences) are under different rates of selection?Are any sites identified to be under diversifying selection within a toxin having a possible structural or chemical impact on the toxin?

To answer these questions, we are using a combination of selection tests from the Datamonkey Adaptive Evolution Server FUBAR (Fast Unconstrained Bayesian Approximation, MEME (Mixed Effects Model of Evolution) and aBSREL (Adaptive Branch-Site Random Effects Likelihood), CodeML from PAML and TreeSAAP (Tree Selection on Amino Acid Properties). Using these in combination will enable us not only to describe the selection rates in mammals but begin to discuss the possible function of a toxin or selection pressures acting upon it due to the different layers of selection we are looking at. For question one, CodeML can provide the overall dN/dS ratio—comparing if the models are significant will identify the overall rate of selection acting upon that toxin.

For question two, using the DataMonkey Adaptive Evolutionary Server, each toxin group alignment will generate a phylogenetic tree to estimate the rate of dN/dS along a branch or at a specific site; FUBAR identifies pervasive site selection (found at a site across the whole phylogeny generated), MEME identifies episodic site selection (found at a site across a subsection of the phylogeny generated) and aBSREL identifies if sites along a branch within the phylogeny has undergone episodic selection. These methods in combination can help identify how much variation there is within sequences and where in these toxins groups there exists variation (especially if highlighted by episodic method like MEME and aBSREL), which may allow toxins to work on different targets. For example, the venom of shrews is predominately comprised of KLK1 but shrews consume both invertebrate and vertebrate prey [17].

Finally, to investigate question three, we will use TreeSAAP that calculates the likelihood of diversifying selection at a site and verifies if that selection could impact an amino acid property (such as hydropathy, the measure of how hydrophobic or hydrophilic a molecule is). Results from TreeSAAP can implicate if selection pressures are having an impact on the toxicity of the molecule and in turn, contribute significantly to the fitness of the mammal.

Although sequence data have improved for mammal toxins, there is still limited availability of the groups we can investigate as well as uncertain identification of non-toxin precursors. Due to this, the toxin groups we are using do not have non-toxins sequences for comparison. Therefore, we cannot state *categorically* if selection identified within this study has been selected for toxicity, but we do know that sequences used in this research have already identified to be recruited as toxins and therefore the possibility of selected sites being implicated in toxicity is still plausible. The methods chosen not only allow a complete comparison of all toxin groups identified within venomous mammals, but enable mammals to be compared to other venomous taxa: three of the above programmes (FUBAR, MEME and CodeML) are used in Sungar and Moran’s paper [9], three are used by Barua, et al., [39] (aBSREL, MEME and CodeML) and four by Low et al., [23] (FUBAR, MEME, CodeML and TreeSAAP). For further justification and explanation of each method used, see Section 5 (Materials and Methods).

Based on previous research, we hypothesize that the overall rate of evolution within mammals will be acting under diversifying selection [16,17,23], and be linked to the evolutionary age of venomous mammals and the different functional usages of venom. We expect to see a proportional amount of diversifying selected sites found across all toxins, although we expect to see more episodic diversifying selection than pervasive selection in KLK1 due to the diverse diet that Eulipotyphla consume requiring more targets to be hit when compared to other toxins which are more focused on smaller targets (i.e., intraspecific in platypuses or blood feeding in vampire bats). Finally, we expect to see some selected sites found on active sites or other structurally important areas within the toxins’ protein, particularly in ‘younger’ groups of mammals (i.e., Desmodontinae) when compared to ‘older’ groups of mammals (i.e., Eulipotyphla).

## 2. Results

For a toxin group to be analysed, at least four sequences representing that toxin had to be identified to minimise errors and skew. We identified eight venomous mammal species with a total of 139 toxin (or non-toxin but found within the venom) sequences that have been described. Due to the criteria, two species that had sequence data (*Blarinella quadraticauda* and *Nycticebus javanicus*) were excluded as each had less than four sequences. Of the 139 sequences identified, only 82 representing seven different toxin groups were able to be used: Plasminogen Activator (Chiroptera), KLK1 (*Blarina brevicauda*, *Solenodon paradoxus*, and *Desmodus rotundus*)), CRiSP (Cysteine-rich secretory proteins, *Desmodus* only and *Desmodus* and *Ornithorhynchus anatinus*), PACAP (Pituitary Adenylate Cyclase-Activating Polypeptide Pituitary Adenylate Cyclase-Activating Polypeptide, *Desmodus*), Kunitz Domain One (*Desmodus*), Kunitz Domain Two (*Desmodus*) and Desmallipins (*Desmodus* allergen-related lipocalins, *Desmodus.*) We also chose to conduct species or order specific alignments (in the case of CRiSP and KLK1), which brought the total number of toxin alignments up to eleven (Appendix A). The majority of alignments (seven out of eleven) are only represented by single species, evolutionary studies on mammals [23] have shown that the methodology still provides informative answers although caution should be taken regarding inferring toxicity and false positives. No sequences could be obtained from slow loris species. This is due to both a lack of sequenced venom and genomes assembled to chromosome level, either of which would have enabled sequences to be included.

### 2.1. HyPhy DataMonkey

#### 2.1.1. FUBAR and MEME Results

Overall trends reported by FUBAR and MEME were consistent across both analyses on the number of sites evolving under purifying or diversifying selection for each alignment, with equivalent numbers of MEME diversifying sites found to as many FUBAR diversifying sites. (Table 2). All but a few sites identified by FUBAR and MEME are different from each other (as they are testing different aspects) although there are some sites identified on alignments that are shared (see Figure 2, point 179). Six alignments have less than or exactly ten sites per method identified to be evolving under either diversifying or purifying selection: PACAP, Desmodus CRiSP, and Kunitz Domain Two. Two alignments have more than ten sites estimated to be evolving under diversifying selection by one or both methods: Plasminogen Activator (FUBAR identified 15 sites evolving under diversifying selection) and Desmallipins (FUBAR identified 22 sites and MEME identified 15 sites evolving under diversifying selection). The four KLK1 alignments have a significant number of sites identified to be acting under purifying selection by FUBAR (Blarina KLK1 *n* = 23, Blarina KLK1 *n* = 16, Eulipotyphla KLK1 *n* = 35 and All KLK1 = 47). However, MEME identifies a large amount of diversifying episodic selected sites in Eulipotyphla KLK1 (*n* = 16) and All KLK1 (*n* = 21) despite also reporting that both alignments are evolving under purifying selection (Eulipotyphla KLK1 ω = 0.593 and All KLK1 ω = 0.513) (Figure 2). This result is likely due to the convergent origin of KLK1 within Eulipotyphla and Chiroptera. As well, in both CRiSP alignments, MEME did not identify any episodic diversifying selection.

#### 2.1.2. aBSREL Results

Diversifying episodic selection was identified on nine of the alignments. Four alignments had one branch identified as evolving under diversifying selection (Blarina KLK1, Desmallipins and Kunitz Domain One). Five alignments had multiple branches identified to evolve under diversifying selection including Plasminogen Activator (three branches), Solenodon KLK1 (five branches), All KLK1 (five branches), PACAP (five branches) and Eulipotyphla KLK1 (six branches). No evidence of episodic selection was found by aBSREL in three of the alignments: Desmodus CRiSP, CRiSP All and Kunitz Domain Two (Appendix A).

### 2.2. PAML Selection Results

#### 2.2.1. Diversifying Selection Detected by Alternative Model 

Reporting the result taken from the M8 model, most mammal toxins are evolving under diversifying selection, with high levels of significance (*p* < 0.001) between all comparisons of null and alternative models (M0 vs. M3, M1a vs. M2a, M7 vs. M8 and M8 vs. M8a) confirming that the alternative model (those that allow for ω > 1) should be accepted. We accept significance as *p* < 0.01 for PAML results to reduce the likelihood of a Type 1 error. Toxins confirmed to have high levels of diversifying selection included Plasminogen Activator (Model 8 ω = 1.3082) (Appendix A), Desmallipins (Model 8 ω = 1.3393) (Appendix A), PACAP (Model 8 ω = 1.3208) (Appendix A) and CRiSP Desmodus only (Model 8 ω = 2.0806) (Appendix A). CRiSP All received a lower but still significant (*p* < 0.01) level of support for M1a vs. M2a and M7 vs. M8 but had significance *p* < 0.001 for an acceptance of the alternative model for M8a vs. M8, indicating diversifying selection (Model 8 ω = 1.2923) (Appendix A).

#### 2.2.2. Purifying Selection Detected by Alternative Model 

All four of the KLK1 alignments are evolving under purifying selection—even with the acceptance of the alternative model with high level of significance (*p* < 0.001) across all four sets of models: KLK1 Blarina (Model 8 ω = 0.8075), KLK1 Solenodon (Model 8 ω = 0.7073) and Kallikren-1 Eulipotyphla (Model 8 ω = 0.6260). Kallikren-1 All also is highly significant (*p* < 0.001) for three of the four sets of models, except for M1a vs. M2a, which has a high level of significance (*p* < 0.01), still indicating purifying selection (Model 8 ω = 0.5592) (Appendix A).

#### 2.2.3. Alternative Models Rejected 

The two Kunitz Domain groups provided mixed results of significance—Kunitz Domain Two is highly significant for the acceptance of alternative model M8a vs. M8 (Model 8 ω = 2.6283); with a significant (*p* < 0.01), result for the acceptance of M1a vs. M2a and M7 vs. M8. Although the M0 vs. M3 acceptance rate rejects the alternative model (*p* < 0.05), the null model still proposes that the group is evolving under a diversifying selection rate (Model 0 ω = 2.8040) (Appendix A). Kunitz Domain One has a very high level of acceptance for the M0 vs. M3 alternative model (ωM3 = 1.4033) but poor levels of acceptance (*p* < 0.05) for the other model matches, therefore, rejecting the alternative model (Appendix A).

### 2.3. Diversifying Sites Detected by Model 8 BEB

The Bayes Empirical Bayes was implemented in Model 8, which detected the sites in each alignment that are evolving under a very high (0.95–0.99) level of diversifying selection. A total of seven alignments had below 10% of total sites (taken from trimmed result in CodeML) evolving under detected diversifying selection: Solenodon KLK1 (3%, 6/187 total sites), Eulipotyphla KLK1 (3%, 5/179 total sites), All KLK1 (3%, 4/151 total sites), PACAP (3%, 6/182 total sites), All CRiSP (6%, 7/122 total sites), Blarina KLK1 (6%, 16/269 total sites) and Plasminogen Activator (8%, 24/303 total sites). Three of the alignments had above 10% of all sites evolving under diversifying selection: CRiSP Desmodus (11%, 14/124 total sites), Kunitz Domain Two (11%, 7/61 total sites) and Desmallipins (17%, 29/167 total sites). Due to the rejection of the alternative Model 8, the number of positive sites reported for Kunitz Domain One (9%, 4/47 total sites) are not included.

### 2.4. TreeSAAP Results 

#### 2.4.1. Properties Detected in Alignments by TreeSAAP

All but one alignment (PACAP) was identified to have sites evolving under diversifying selection that impacted different amino acid properties. The properties identified to have been affected across the alignments are long range non-bonded energy (detected in one alignment, CRiSP Desmodus), normalised partial specific volume (detected in one alignment, Kunitz Domain Two), consensus hydrophobicity (detected in one alignments, Kunitz Domain Two), power to be at the middle of the alpha-helix (detected in two alignments, Solenodon KLK1 and KLK1 All), polar requirement (detected in two alignments, CRiSP Desmodus and All CRiSP), hydropathy (detected in two alignments, KLK1 Eulipotyphla and KLK1 ALL), power to be at the C-terminal (detected in four alignments, KLK1 Shrew, KLK1 Eulipotyphla, KLK1 All and CRiSP All) and isoelectric point (detected in five alignments, Plasminogen Activator, KLK1 Shrew, Desmallipins, CRiSP All, Kunitz Domain One) (Appendix A).

#### 2.4.2. Alignments Which Had Significant Impact on Properties Identified by TreeSAAP

Only eight of the eleven alignments had specific sites identified by TreeSAAP: KLK1 Shrew, KLK1 Solenodon, KLK1 Eulipotyphla, KLK1 All, Desmallipins, CRiSP Desmodus, CRiSP All, and Kunitz Domain 2. Further, the four KLK1 alignments were identified to have more than 1 site present for each property identified. Some sites were also recorded to be evolving at diversifying selected rates that affected amino acid properties, however they were recorded at a different magnitude than observed.

#### 2.4.3. Location of Selection on the Protein Structure

When mapped to the protein structures, the positions under diversifying selection identified by multiple methods were located on the proteins’ surfaces. Based on functional annotation, no active site residue (or region) is affected. In the plasminogen activator, one glycosylation position is under diversifying selection, as is an initial position in the signal peptide region of PACAP (Figure 3).

## 3. Discussion

Using different branch and site-specific methodologies, as well as TreeSAAP, we analysed the rate of selection acting on toxins found across venomous mammals for the first time. Overall, mammal toxins are evolving under diversifying selection (Figure 4). Six are evolving under diversifying selection (Plasminogen Activator, CRiSP, Desmallipins, PACAP, Kunitz Domain Two and Kunitz Domain One (although Kunitz Domain One is not a significant result). In contrast, the KLK1 toxin group (including all species-specific alignments) is under purifying selection. These results indicate that mammalian toxins are overall evolving under diversifying selection, following the two-speed model. There are 89 diversifying sites (>0.95) identified by PAML Model 8 across the toxin groups, with the majority from Plasminogen Activator (*n* = 24) and Desmallipins (*n* = 29). The total dN/dS ratio of mammal toxins (12.3% dN, 85.7% dS) and large number of diversifying sites identified would group mammal toxins with the other high synonymous groups found by Sungar and Moran [9], the advanced snakes (diversifying sites = 531, diverged ~54 MYA) and cone snails (diversifying sites = 166, diverged ~33–50 MYA).

Throughout all the toxins, there is evidence of similar amounts of pervasive and episodic selection found in each alignment. While this is observed in a number of other venomous orders [9], the very high amount of episodic selection identified in KLK1 Eulipotyphla and KLK1 All (by both MEME and aBSREL) and the lack of episodic selection identified in both CRiSP alignments are both. This episodic selection identified in KLK1 could be to do with the diverse diet that venomous Eulipotyphla species are observed to eat [16,17,20]—with multiple KLK1s able to target both physiologically different invertebrate and vertebrate systems at the same time [16,17,20]. When observed, *Neomys* spp., show a preference for taking large prey items (such as frogs or rodents) over smaller prey items [19] and venom of *B. brevicauda* has been shown to be potent on vertebrates as large as domestic cats (*Felis catus*) [13] despite the shrew’s size—it is possible that this episodic selection has slowly developed as shrews expanded their niche from to include vertebrates with selection pressure and competition from other invertebrate-eating organisms pushing them to target larger, risky but more rewarding prey.

Structural models of mammalian toxins revealed that positions under diversifying selection are located on protein surfaces (Figure 3). This is in accordance with the observation that changes in exposed residues are less detrimental to protein stability than those occurring in buried residues [40,41,42]. These changes in surface-exposed residues are considered facilitators for neofunctionalisation of toxins by modification of interactions between protein and its target(s) [2,43]. For KLK1, the active and glycosylation sites are not under selection. Positions under selection are located in regions close to regulatory loops identified as critical for toxicity in KLKs from reptiles and mammals [44]. For plasminogen activator, the active site is not under selection, but one glycosylation site is. Glycosylation is relevant for protein secretion and localisation [45], something that has been shown to be critical for some vampire bat toxins [46]. In PACAP, the identified residue under selection is located in the N-terminus of its signal peptide segment [47]. Signal peptides are ubiquitous in animal venom toxins [43], being essential for protein secretion [47]. For Desmallipins and Kunitz Domain Two, there are no pinpointed functional positions, with all sites under selection being found in the protein surface. The selection pressure on positions related to protein secretion and localisation may point to a progressive transition of these proteins to be present in venom glands (or in alternative glands/sites). Likewise, the pressure on surface coils on KLK1 may indicate further target alteration (either broadening prey classes or becoming more active towards specific groups). We hypothesised that we see more significant sites under selection on evolutionary younger groups; while the results do concur with that, this is more likely because sequence data are overwhelming from *Desmodus rotundus*.

Research has already indicated that toxins within mammal venom are experiencing mixed selection rates [9,16,17,23,24]. In Chiroptera, the majority of toxins are evolving under weak to strong diversifying selection [23] with an unusually high proportion of sites evolving under diversifying selection (from 20 to 36% of total sites). Within Eulipotyphla, the *Solenodon* KLK1 paralogs are evolving under purifying selection compared to other mammal non-toxic KLK1 despite a number of diversifying sites identified within the alignments [16] but in *Blarina*, KLK1 toxins are identified to be evolving under diversifying selection compared to non-toxin KLK1 paralogs [17]. Although these results are highly credible and provide support for the two-speed model, there has been little consistency with the methodology used due to the advancements made in selection tests since 2001 and the varied areas of research by each study. This mixed pattern was also detected by Harris and Arbuckle [48] when examining loss/gain of venom (and poison) within tetrapod orders, with mammals displaying a higher rate of losing venom as a trait despite only venom gains being recorded by ancestral shifts. The authors explained that this could be a result of venom in mammals evolving as an ancestral trait but being lost with no trace. Alternatively, due to the unusual spread of the trait and the multiple specific usages of venom recorded in the mammals, venom as a trait may not have evolved at the same time as the divergence of venomous mammals. Therefore, toxins may still be undergoing high levels of diversifying selection due to the more recent recruitment of venom within venomous mammals. Our results do corroborate previous work, with the general trend of whether a toxin is acting under diversifying or purifying selection. There are differences between the proportion of sites under significant selection and the site numbers identified. This could likely be due to differences in alignment, using a different methodology or the inclusion of different sequences compared to past research on mammal toxins.

### 3.1. Mammal Toxins Are Acting under Diversifying Selection

We confirm previous suggestions [16,17,23] that the majority of mammal toxins are evolving under diversifying selection. Taking into account the evolutionary ages of each group of mammal analysed (~178 MYA for Monotremes, ~73 MYA for Eulipotyphla and ~5 MYA for Chiroptera [12,16,34,49]), mammal toxins fit the two-speed model as a ‘young’ evolutionary group.

The specific molecular mechanism responsible for toxins may contribute to the diversifying rate of selection observed in this study; while duplication were thought to be most responsible for toxins in venom [7,50], there are other mechanisms that also identified in the neofunctionalisation of toxins [51]. For example, alternative splicing (where exons from the same gene are spliced in different mRNA strings to produce differing proteins) is responsible for the different plasminogen activators identified in all three species of vampire bat, a mechanism that can accelerate selection rates [52].

Our findings support previous research that suggested KLK1 occurs under purifying selection [16]. As this toxin predominantly has a predatory function (See Figure 1), more research is required to investigate why this does not have higher selection pressure as would be expected in the predator–prey arms race [17,18,53]. Our results for venom of *Blarina* indicated that other molecules, such as HYLAP (Hyaluronidase acid, PH-20), may play a role as they facilitate the ‘spread’ of venom by degradation of target cells. These molecules are under diversifying selection [17] and may act to maintain KLK1 under purifying selection by minimising mutations. Given, HYLAP was not identified in all KLK1 venoms (such as Solenodon) further research is required to identify equivalent molecules in those venoms.

Another intriguing aspect of the purifying selection KLK1 is under is the number of diversifying sites identified by PAML Model 8 in *B. brevicauda*—a total of 16 sites (Appendix A). Although this may appear to be a false positive (indeed, FUBAR identify significantly more pervasive purifying than diversifying sites across the *B. brevicauda* KLK1 alignment, Appendix A), in [44], *B. brevicauda* toxic KLK1 was identified to be under purifying selection but inserted loops were evolving under episodic diversifying selection, raising the suggestion that these sites are responsible for the toxic effect by these molecules. Indeed, all KLK1 alignments had diversifying sites that TreeSAAP identified as having affected a structural protein property (Appendix A). Research on KLK1 proposed it as a toxipotent molecule [39]—that even prior to neofunctionalisation of KLK1 in venom, it possessed properties that made it suitable for venom usage. Due to the parallel origins of KLK1 toxins identified in mammal and reptile venom, sharing structural and functional similarities despite the divergence of the groups 300 MYA, KLK1 was suggested to be more susceptible to becoming a toxin than other related groups [54]. This would explain the intense pervasive purifying selection acting on KLK1, with its ancient origins but with specific episodically selected sites that are toxipotent. Although *S. paradoxus* and the grouped KLK1 that included all identified KLK1 in the mammals had significantly less positively sites (only six and four respectively), this could be a result of the convergence origin of KLK1.

### 3.2. Different Functions Have Different Rates of Selection on Mammal Toxins

In the Class Mammalia, venom does appear to follow the two-speed model but when examining the separate mammalian orders in this study (Monotremata, Eulipotyphla and Chiroptera), we found no clear pattern, even when considering site or branch selection rates. Beyond the evolutionary divergence times of each order, it is the functional usages (even when broadly grouped under predatory, defensive, or intraspecific), ecology and behaviour of the animal that appear to have a stronger influence on the rate of selection of a particular toxin.

Only three of the toxin groups identified (KLK1, Desmallipins and CRiSP) that have diversifying sites identified by TreeSAAP as significantly affecting the toxins’ protein properties and therefore have a significant impact on the fitness of the animal. These results differ from those reported by Low et al., [23] who reported that Kunitz Domain One, Kunitz Domain Two and Plasminogen Activator had significant sites reported by TreeSAAP. This is likely due to the differences in methodology, as we used nucleotide sequences whereas they used amino acid sequences.

In other toxin groups (except for PACAP), diversifying sites that had influenced nucleotide sequences were identified although they were not significant. A possible explanation for why only these three groups had significant sites in TreeSAAP could be due to pluripotency of toxins—Desmallipins are allergen-related lipocalin proteins, a candidate for olfactory communication in *D. rotundus* [55], a social species that partakes frequently in grooming. This sociality could also explain the high number of diversifying sites identified with both FUBAR and MEME—in particularly the episodic selection indicating different levels of diversifying selection acting on 22 sites within Desmallipins. The data used within this study come from two separate studies on *D. rotundus* [23,56]—if sociality is a selection pressure on Desmallipins, this could be why there are so many sites, as two different colonies of bats will have distinct communication between each other. This is similar to slow lorises, which use brachial gland exudate (BGE), a toxin with high structural and functional homology with Fel-d-1 but which is still involved in olfactory communication [25]. For the CRiSPs, both in *D. rotundus* and *Ornithorhynchus anatinus*, they might preform a role in immunity—although vasodilation (as proposed in [23]) may be a stronger factor for *D. rotundus*. Further expanding on the CRiSP pleiotropy, the PAML Model 8 result for CRiSP All (M8 = 1.2923) while still weakly diversifying, is significantly less than just CRiSP *Desmodus* (M8 = 2.0806). Running the different CRiSP sequences through GenBank BLAST [57] shows that the CRiSPs in *O. anatinus* and *D. rotundus* evolved from different paralogs, with *O. anatinus* evolving from CRiSP 2 and *D. rotundus* evolving from CRiSP 3. The non-toxic mammalian CRiSPs are expanding under purifying selection (although both [58,59] identify that CRiSP 2 is not-significant) and while both are associated with immunity, CRiSP 2 is recognised for its role in sperm gamete fusion. Further evidence to support this is the fact that no episodic selection was identified in either CRiSP alignment, this could imply that the function for each is still somewhat retained as there is little to no non-synonymous mutation between each sequence identified. The *O. anatinus* CRiSP is also expressed at the same level within the venom gland both during and out of breeding season, further providing evidence of a possible pleiotropic nature [38,60].

The higher rate of selection on Chiroptera toxins can also be interpreted as genomic evolution. The jump from the ancestral diet of insectivory to blood feeding in the vampire bats occurred approx. 5 MYA [61]. Haematophagy requires numerous adaptions due to the potential dangers of viral infection, iron poisoning and poor quality of nutrients [52,62]. In *D. rotundus*, adaptions to aid in this diet include a heavily adapted microbiome that has evolved parallel to the bats genome and a unique immune system amongst mammals, with loss of immune genes that result in an inflammatory response to viruses (include defensins and natural killer) but had an expansion in major histone complex genes that result in less cell death [63]. The genome of *D. rotundus* also showed upwards of 1.6–2.6 times higher number of transposon mutagenic changes compared to other Chiroptera, with these transposons found in regions related to immunity amongst other adaptions related to haematophagy [62]. This could have resulted in rapid selection rate in vampire bats on homeostasis genes and immune genes, as well as genes associated with secretion [64], in comparison to other Chiroptera species. Vampire bats regurgitate blood to other bats within a colony if they were unsuccessful in feeding, with gut microbiomes closer in similarity between bats that had engaged in food sharing compared to kin or social groups [65]. Whilst many of the toxins found in vampire bat venom are posited to aid in blood feeding at the host (via anticoagulation or vasodilation [56]) and could explain the higher rate of evolution due to predator–prey interactions, the constant threat of viral infection could also justify why some of the toxins in vampire bats (i.e., CRiSP or Desmallipins) are under intense diversifying selection pressures.

Whilst PACAP is evolving under moderate diversifying selection (M8 = 1.3208) and has 14 diversifying sites identified, it was the only toxin that had no properties identified by TreeSAAP. Previous work suggested that this indicates the rate of selection and positive sites therefore had minimal impact on the fitness of *D. rotundus.* The discovery of vCGRP (vampire bat-derived form of calcitonin gene-related peptide) [33] which causes intense vasodilatory properties indicates that multiple toxins found within venom of *D. rotundus* are responsible for similar functions. Like the mixed result from the two Kunitz Domains (with a strong diversifying selection acting on Kunitz Domain One and weaker diversifying but not significant selection acting on Kunitz Domain Two), which have a proposed anticoagulation effect similar to plasminogen activator, having multiple toxins with similar functions can lead to relaxed selection.

A final point worth considering is the co-evolution of toxins with their target and the toxin families that represent several groups within this study. As observed by Zhu, et al., [66], some homologous scorpion alpha-toxins in *Mesobuthus* spp. contain insertions that reduce the efficiency of an alpha-toxin on mammalian voltage gated Na+ channels. These insertions did increase the insecticidal effects of the alpha-toxins, an adaption that allows the scorpions to take a greater variety of prey more efficiently and that prey selection had been a selection pressure on the alpha-toxins [67]. Toxins groups used in this study came from species-specific toxin families (i.e., KLK1 Solenodon), with aBS (Appendix A) indicating that some branches have had diversifying selection acting on them at some point (i.e., KLK1 9643 and KLK1 7098)—these different rates of selection and multigene copies of the same toxin may allow venomous mammals to take different prey (such as the case of the solenodon that consumes both vertebrate and invertebrate prey) or keep the ancestral function of a toxin (as discussed above with CRiSP and Desmallipins). This would require further testing on a molecular, protein or genomic level of venomous mammals and their targets to verify this.

### 3.3. Future Research 

The results are overwhelmingly represented by one species (*Desmodus rotundus*), which accounts for six of the seven toxin groups examined here. Additionally, the toxins analysed in this study have a small sample size (ranging from four to 16); smaller sample sizes are more likely to be skewed by sequences that are acting under very strong selection rates. To mitigate this, steps were taken to maximise the stability of the results with high levels of bootstrapping or taking only very significant results for CodeML (defined as *p* < 0.01). We cannot exclude the possibility of skewed or insufficient data within this paper. Therefore, caution should be taken when interpreting these results—although numerous results concurring with previous work do aid in providing support to our results.

The lack of primate toxin sequences also limits this research. Currently, only two amino acid and one nucleotide BGE sequence have been identified; all three sequences were identified by mass-spectrometry [26]. The slow lorises are the only example of a two-step venom system (with the BGE gland needing to be mixed with saliva via licking to induce potency) [11]. Identifying any additional molecules found in the gland for comparison and collecting a full sequence of BGE from multiple slow loris species or specimens would significantly increase our understanding of mammalian venoms. As well, intraspecific behaviour between slow lorises is territorial, between male-male and female-female [30,32]. Venom composition, toxicity and amount differs between the sexes in some venomous species [1]. Identifying if the rate of selection acts similar on venom within each sex in slow lorises would be an excellent contribution to our understanding of venom systems.

Identifying non-toxin precursors would allow researchers an opportunity to infer which sites or branches have been selected for in the transition to toxicity. Using the gene-wide methods such as BUSTED (Branch-Site Unrestricted Statistical Test for Episodic Diversification) to seek out where episodic positive selection has occurred across a gene (or a subset of branches) or RELAX which identifies if rates of purifying selection differ across a tree (this method requires a test set of branches, which would be non-toxin homologs). Identifying which specific non-toxic precursor (as well as any transitional non-toxin, such as venom like beta-defensin (DEFB-VL) found in the platypus [14]) has led to a mammalian toxin would provide a fascinating opportunity to identify what has driven parallel recruitment of similar toxins as mentioned in the Discussion about CRiSP recruitment between Chiroptera and Monotremes. Further expanding upon this point would provide excellent evolutionary insight.

There are additional shared toxins in mammal venoms that are widely shared by other venomous animals including phospholipase A2, secretoglobin and acid hyaluronidase. If further sequences are discovered in species not yet sequenced (i.e., *Crocidura canariensis*) [22] or species are revised for better quality sequences, they would provide an excellent comparison within the class and also across different animal venoms.

## 4. Conclusions

For the first time, the selection rates of available mammal toxins have been analysed together. We have shed light on the evolution of venom within mammals, reviewing cutting edge ecological, behavioural, and genomic research, as well as previously discovered toxins, to compare mammals to more well studied taxa. Overall, mammal toxins are acting under a diversifying selection of evolution and have a significantly large number of sites across eight toxin groups that are particularly evolving under both pervasive and episodic diversifying selection which does follow the two-speed evolutionary model. These results can be indicative of different selection pressures acting on venomous mammals including sexual selection, immune responses, pluripotent proteins balancing toxic and non-toxic roles and predisposition of certain proteins for toxicity. Several toxins identified (KLK1, Desmallipins and CRiSP) have positively selected sites identified by TreeSAAP and visualised as protein models implying that these are significant to the fitness of the animal. These sample sizes contain few sequences: obtaining more samples of described toxins from numerous mammals (such as the platypus specific OvDLPs) and obtaining high quality sequences from missing toxins (e.g., BGE from *Nycticebus* or toxins from *Crocidura canariensis*) would provide more conclusive results and better support for understanding which factors are greater selection pressures on toxins in different mammal groups. It is only the beginning of furthering our understanding of this fascinating venomous group and the applications for its venom.

## 5. Materials and Methods

### 5.1. Data Collection 

Data collection methodology and criteria is identical to the methodology as used by Sunagar and Moran, 2015 [9]. There is speculation on which species of mammal are venomous; we decided to be conservative in our definition and only included species that had published, accessible complete nucleotide sequences available for use. Thus, data from *Sorex araneus* and *Neomys foidens* were not included as venom has only been identified as fractions by Mass Spectrometry. Very few molecules from mammal toxins have been isolated and their toxicity verified—we assume unless specified that molecules described in text by researchers as toxic based on homology or experiments are toxins (including Plasminogen Activator [23,24] and Kunitz [36]). Identifying selection on mammal toxins within an alignment requires that there are at least four sequences represented by that toxin available—these criteria excluded the majority of available toxins identified in mammals as only one sequence has been published. Data were predominately collected from published literature using either review papers to identify recently published papers, then using key word searches on Google Scholar (‘venomous mammal’, ‘poisonous mammal’ or ‘toxins mammals’) and then finally cross referencing the found toxin sequences using UniProt, GenBank and EMBL-EBI to confirm best available sequences for analysis. To ensure no sequences where missed, a final search on GenBank and UniProt using either the venomous species or toxin name were conducted. Sequences were also obtained via individual repositories or via emailing researchers (pers. comm. Nick Casewell). For a full list of where sequences were obtained, see Appendix A.

### 5.2. Alignments and Trimming

Toxins that had multiple representatives were identified and their nucleotide sequences where grouped together in Geneious Prime v2022.1.1 (https://www.geneious.com/; Biomatters, Auckland, New Zealand) (Appendix A), aligned with MUSCLE 3.8.425 [68] and finished with manual editing of alignment including necessary removal of STOP codons for use in analysis. Genus or Order specific alignments (such as CRiSP *Desmodus*) were done as well as Class wide alignments to confirm significance as well as identify site-specific changes.

Aligned sequences were run through TrimAI v1.3 (accessed via Phylemon 2.0 web server) [69] on automated1 mode (chosen as this mode is optimised for alignments used in maximum likelihood analysis and identifies if gappyout or strict will be more suitable to use on the alignment based on average identify score, sequence number and maximum identity score). Spurious sequences were not removed. Trimmed output was downloaded as FASTA files.

### 5.3. Datamonkey Selection Analysis 

Selection analysis was conducted on Datamonkey Adaptive Evolution Server [70,71,72] looking at both branch-site and site-specific selection levels on each alignment. Site-specific models examine which amino acid positions in an alignment are evolving under selection, with level of significance given at 0.95 or 0.99. Site-specific models cannot indicate which sequences the site-specific selection is found in. Branch site level selection levels can indicate which branch or node selection has occurred in an alignment and specifically which sites.

### 5.4. Site-Specific Selection 

#### 5.4.1. Fast Unconstrained Bayesian AppRoximation

FUBAR (Fast Unconstrained Bayesian AppRoximation [73], a Bayesian selection model) was used to examine pervasive site selection. FUBAR assumed a constantly level of selection across the whole alignment and significance is taken at posterior probability < 0.9. FUBAR reports if diversifying or purifying selection is present at a site.

#### 5.4.2. Mixed Effects of Model Evolution

MEME (Mixed Effects of Model Evolution [74], a maximum likelihood model) was used to examine episodic site selection with significance taken at <0.05. MEME will examine a proportion of branches at a time and then infers two different ω classes with different weighting, Null and Alternative. Both classes measure alpha (dS) and two different beta (dN) values, Beta+ and Beta-. Null allows values in alpha to be unrestricted but both beta values must be smaller or the same as alpha. Alt allows values in alpha and beta+ to be unrestricted but beta- must be the same or smaller than alpha. Sites are examined for the probability that a site evolves at each ω class on a given branch. If beta+ is greater than alpha (and adjusted with LRT) then a site is assumed to be under diversifying selection. MEME will report if you have diversifying selection present at a site only.

### 5.5. Branch Specific Selection

#### Adaptive Branch-Site Random Effective Likelihood

aBSREL (Adaptive Branch-Site Random Effective Likelihood [75]) was used to examine episodic branch-site selection on each alignment with significance set at <0.05 and all branches selected. aBSREL measures both site and branch ω values but will only report branch results for tests. aBSREL examines each branch in the phylogenetic tree it constructs from the alignment to measure which proportion has evolved under diversifying selection. The Akaike Information Criterion corrected (AICc) is used to calculate the optimal rate present on each branch and LRT is used to fit the best model on each branch. These optimal models are then compared to a null mode to identify diversifying selection. aBSERL will report if you have diversifying selection present on a branch.

### 5.6. PAML Selection Tests

#### 5.6.1. Tree Data

Maximum likelihood trees were generated for each trimmed alignment, with an appropriate model selected using MEGAX [76] model selection test (Appendix A), ran with 1000 bootstrap replicates and unrooted. Each tree was then saved as two separate Newick files (one with branch lengths and support, one clean tree with no labels).

#### 5.6.2. Selection Tests Using CodeML

Trimmed FASTA files were first converted into PAML files using the file converter function available in EasycodeML [77]. For the selection tests, CodeML (from PAML v4.8 [78]) was opened via the command line. To confirm if the dN/dS site-specific results are significant, the alternative model must be compared to a null model and compared via an LRT. Therefore, six models were chosen for site-specific comparison: three null models, M0 (assumes constant ω value across alignment), M1a (assumes only neutral or purifying evolution across the alignment) and M7 (assumes neutral evolution as well but using beta) and three alternative models M3 (allows some level of variation amongst the alignment but very limited) M2a (assumes diversifying, neutral or purifying evolution across the alignment) and M8 (assumes neutral or diversifying evolution as well but using beta)). Additionally, M8a (a null model that assumes neutral evolution using the beta model) was used for comparison with M8 model, as it provides the best supported result for each alignment. The comparisons were M0 vs. M3, M1a vs. M2a, M7 vs. M8 and M8a vs. M8. The clean Newick file and PAML file were placed into a folder with a codeml.ctl file with the above models coded (M8a was coded separately with a new codeml.ctl file as ω had to be changed to equal 1 from default of 0.4). All options remained default excluding the option to clean the data which was selected.

### 5.7. Comparison of Toxin Sequences to Ancestral Group

#### 5.7.1. Tree Selection on Amino Acid Properties

To investigate the impact sites evolving under natural selection have on the protein structure of these alignments and to confer with sites identified by selection tests used in this meta-analysis, TreeSAAP v1.4 [79] was used on each alignment’s FASTA file and cleaned Newick files where converted into NEXUS format and then ran through on an initial ‘length’ sliding window. The Z-score statistics were then used to identify which of the 32 properties were evolving under the highest amount of selection (ranked from 8 (most) to 1 (least)) with only those properties at +3.092 significance and ranked 8–6 taken as the properties worth investigating. The alignments were run again with the list of interesting properties and a sliding window of one (1) was used to examine results.

#### 5.7.2. Visualisation of Results 

Structural models for proteins with amino acid positions under diversifying selection were obtained from AlphaFold Protein Structure Database [80,81]. They were represented as cartoons following Richardson conventions [82] and the images were rendered with PyMol 1.3 (Schrodinger, LLC., New York, NY, USA). Functional annotation was obtained from UniProt [83].

## Figures and Tables

**Figure 1 toxins-14-00842-f001:**
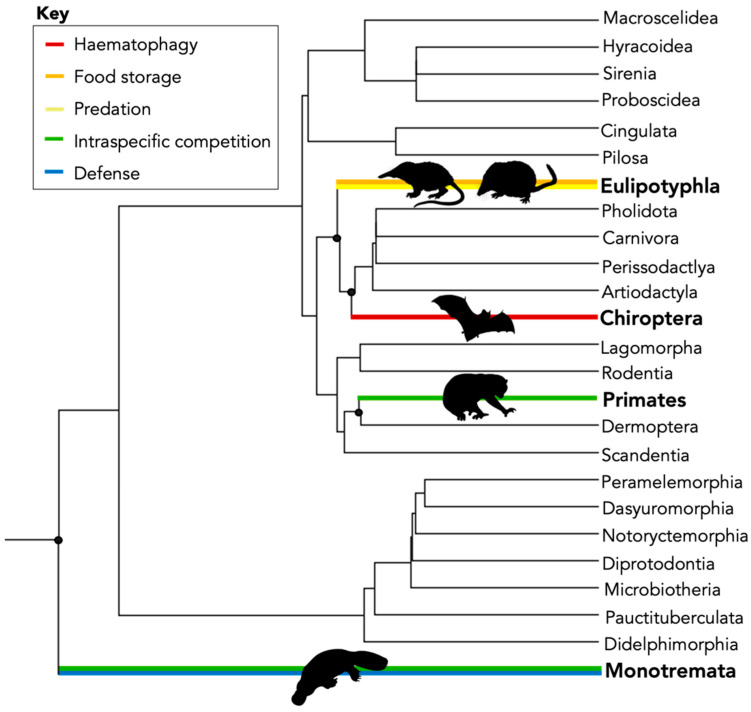
Current phylogeny of the class Mammalia, branches with orders of venomous mammals are coloured with corresponding functional usages of venom. Estimated divergent times for venomous mammals are approximately 178 MYA for Monotremes, 73 MYA for Eulipotyphla, 5 MYA for Chiroptera (Desmodontinae) and 25 MYA for Primates (Lorisidae). Divergent times and proportionally representative branches were obtained from TimeTree [12].

**Figure 2 toxins-14-00842-f002:**
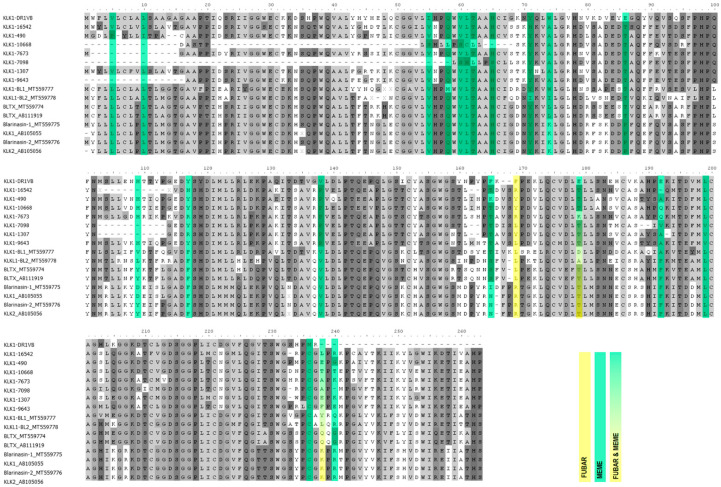
Amino acid alignment of KLK1 All, with diversifying selection highlighted according to each programme. Yellow represents FUBAR (pervasive selection) and Green represents MEME (episodic selection). Overlapping sites with partial or complete programme agreement are coloured as mixed hues. Please see Appendix A for all other alignments.

**Figure 3 toxins-14-00842-f003:**
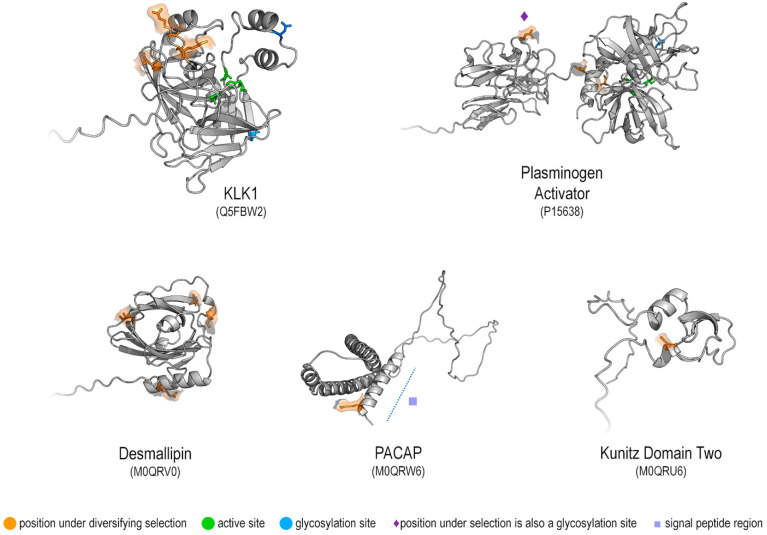
Protein structures highlighting positions under diversifying selection detected in this paper. Overlapping sites with partial (yellow) or complete (orange) FUBAR and MEME agreement are shown as sticks with molecular surface contour on overall cartoon representation. Structure IDs correspond to AlphaFold Protein Structure Database.

**Figure 4 toxins-14-00842-f004:**
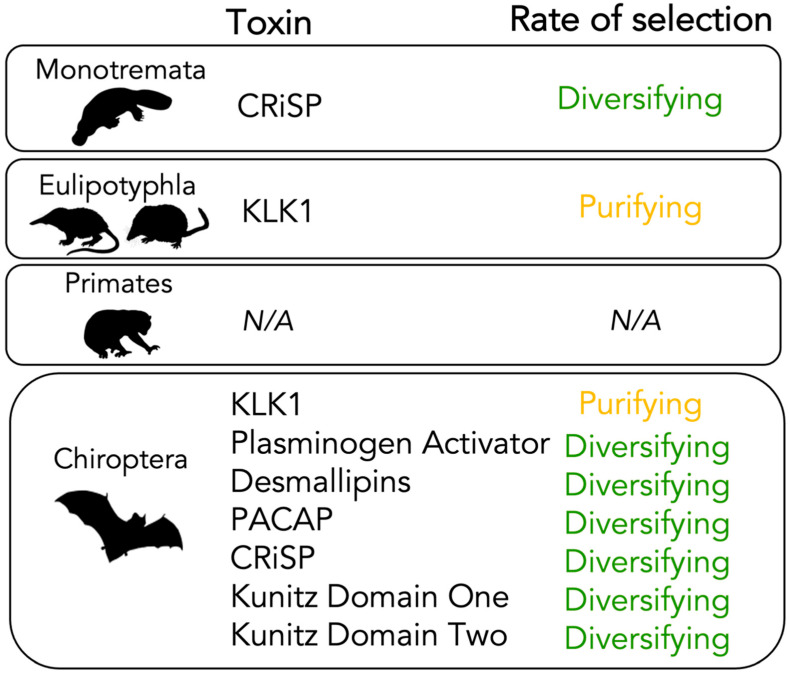
A visual summary of the main findings of our research showing that mammal toxins are evolving overall under diversifying selection.

**Table 1 toxins-14-00842-t001:** The currently confirmed or highly likely venomous mammal species—based on either observation or relation to confirmed venomous species. We have chosen not to include several Eulipotyphla that have been suggested to be venomous as this has been highly speculative for many years with no studies quantifying potential venoms or observing behaviours that confirm likely venom use. For detailed information on specimens of Eulipotyphla that may potentially be venomous, please refer to Kowalski and Rychlik [13].

Common Name	Order	Family	Genus	Species	Venom Status	Reference
Platypus	Monotremata	Ornithorhynchidae	*Ornithorhynchus*	*anatinus*	Confirmed ^1^	[14]
Short-beaked echidna	Monotremata	Tachyglossidae	*Tachyglossus*	*aculeatus*	Vestigial ^2^	[15]

Cuban solenodon	Eulipotyphla	Solenodontidae	*Atopogale*	*cubana*	Highly likely (evolutionary)	[16]
Hispaniolan solenodon	Eulipotyphla	Solenodontidae	*Solenodon*	*paradoxus*	Confirmed	

Northern short-tailed shrew	Eulipotyphla	Soricidae	*Blarina*	*brevicauda*	Confirmed	[17]
Southern short-tailed shrew	Eulipotyphla	Soricidae	*Blarina*	*carolinensis*	Highly likely (evolutionary)	
Elliot’s short-tailed shrew	Eulipotyphla	Soricidae	*Blarina*	*hylophaga*	Highly likely (evolutionary)	
Everglades short-tailed shrew	Eulipotyphla	Soricidae	*Blarina*	*peninsulae*	Highly likely (evolutionary)	

Asiatic short-tailed shrew	Eulipotyphla	Soricidae	*Blarinella*	*quadraticauda*	Confirmed	[18]
Indochinese short-tailed shrew	Eulipotyphla	Soricidae	*Blarinella*	*griselda*	Highly likely (evolutionary)	
Burmese short-tailed shrew	Eulipotyphla	Soricidae	*Blarinella*	*wardi*	Highly likely (evolutionary)	

Mediterranean water shrew	Eulipotyphla	Soricidae	*Neomys*	*anomalus*	Confirmed	[19]
Eurasian water shrew	Eulipotyphla	Soricidae	*Neomys*	*fodiens*	Confirmed	[20]
Transcaucasian water shrew	Eulipotyphla	Soricidae	*Neomys*	*teres*	Highly likely (evolutionary)	

Crawford’s grey shrew	Eulipotyphla	Soricidae	*Notiosorex*	*crawfordi*	Likely (behavioural observation)	[21]
Canarian shrew	Eulipotyphla	Soricidae	*Crocidura*	*canariensis*	Likely (behavioural observation)	[22]

Common vampire bat	Chiroptera	Phyllostomidae	*Desmodus*	*rotundus*	Confirmed	[23]
Hairy-legged vampire bat	Chiroptera	Phyllostomidae	*Diphylla*	*ecaudata*	Confirmed	[24]
White-winged vampire bat	Chiroptera	Phyllostomidae	*Diaemus*	*youngi*	Confirmed	

Pygmy slow loris	Primates	Lorisidae	*Xanthonycticebus*	*pygamaeus*	Confirmed	[25]
Greater slow loris	Primates	Lorisidae	*Nycticebus*	*coucang*	Confirmed	[25]
Javan slow loris	Primates	Lorisidae	*Nycticebus*	*javanicus*	Confirmed	[26]
Kayan slow loris	Primates	Lorisidae	*Nycticebus*	*kayan*	Confirmed	[27]
Bengal slow loris	Primates	Lorisidae	*Nycticebus*	*bengalensis*	Confirmed	[28,29]
Bornean slow loris	Primates	Lorisidae	*Nycticebus*	*borneanus*	Highly likely (evolutionary)	[26,27,30]
Philippine slow loris	Primates	Lorisidae	*Nycticebus*	*menagensis*	Highly likely (evolutionary)	
Sumatran slow loris	Primates	Lorisidae	*Nycticebus*	*hilleri*	Highly likely (evolutionary)	
Bangka slow loris	Primates	Lorisidae	*Nycticebus*	*bancanus*	Highly likely (evolutionary)	

^1^. Male only, active during breeding season (approx. June–October) ^2^. Vestigial gland present, adaptation now possibly for olfactory communication, supports theory that venom is an ancestral trait of Monotremata. See Wong, et al., [15] for further details.

**Table 2 toxins-14-00842-t002:** Summary of results for eleven alignments representing the seven toxin groups (bolded). Results include sites identified by FUBAR evolving under diversifying (ω > 1) or purifying (ω < 1) selection, sites identified to be evolving under diversifying selection by MEME, total dN/dS for Model 0, Model 2a, Model 8 and Model 8a and significant diversifying sites (<0.95–0.99) for Model M2a and M8. Sequence alignments reflecting these results are shown as Appendix A.

Species	Toxin	FUBAR		MEME	CODEML						
		ω > 1	ω < 1	ω > 1	Model 0 Weighted Average	Model 2a Weighted Average	M2a Positive Sites >0.95	M2a Positive Sites >0.99	Model 8 Weighted Average	M8 Positive Sites >0.95	M8 Positive Sites *p* > 0.99
*Blarina brevicauda*	Kallikren-1 (KLK1)	5	23	4	0.57984	0.8139	N/A	N/A	0.8075	13	3
*Solenodon paradoxus*	Kallikren-1 (KLK1)	8	16	6	0.58254	0.7298	1	2	0.7073	2	4
Eupotiphyla (*Blarina* and *Solenodon*)	Kallikren-1 (KLK1)	5	35	16	0.49379	0.6748	1	2	0.6260	3	2
***Desmodus rotundus*, *Blarina* and *Solenodon***	**Kallikren-1 (KLK1)**	**3**	**47**	**21**	**0.43905**	**0.6339**	**1**	**N/A**	**0.5592**	**3**	**1**
***Diphylla ecaudata*, *Desmodus* and *Diameus youngi***	**Plasminogen Activator**	**15**	**8**	**4**	**1.0549**	**1.3046**	**10**	**4**	**1.3082**	**14**	**10**
** *Desmodus* **	**Desmallipin**	**22**	**8**	**15**	**1.09464**	**1.3216**	**3**	**7**	**1.3393**	**17**	**12**
** *Desmodus* **	**PACAP**	**3**	**5**	**2**	**0.90099**	**1.3239**	**4**	**1**	**1.3208**	**14**	**N/A**
*Desmodus*	CRiSP	5	2	0	1.48836	2.0592	4	N/A	2.0806	4	2
***Desmodus* and *Ornithorhynchus anatinus***	**CRiSP**	**7**	**2**	**0**	**0.34629**	**1.3843**	**3**	**N/A**	**1.2923**	**4**	**3**
** *Desmodus* **	**Kunitz Domain One**	**1**	**7**	**1**	**0.29323**	**1.4170 ^NS^**	**1**	**N/A**	**1.3985 ^NS^**	**2**	**2**
** *Desmodus* **	**Kunitz Domain Two**	**2**	**1**	**2**	**2.28040 ^NS^**	**2.6283**	**3**	**1**	**2.6283**	**4**	**3**

## Data Availability

All eleven alignments used for analysis are available to download here. For any additional files or questions from analysis’ used in this study, please contact the first author (L.L.J.F.).

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
