# Peer review of "The Fast and the Furriest: Investigating the Rate of Selection on Mammalian Toxins"

_toxins, 2022, doi:10.3390/toxins14120842_

Round 1

Reviewer 1 Report

The manuscript “The fast and the furriest: investigating the rate of selection on mammalian toxins” is well-written and examines selection pressures across mammalian toxins with a level of detail that has yet to be done, making this a novel study. It is also quite interesting to see the output of multiple selection tests on the same dataset, although some variation is expected with such a small sample size. I feel that with some additional considerations, such as including a larger set of homologous sequences in these analyses and providing figures/visualizations of program outputs, this manuscript would be improved.

Major comments:

1.     It doesn’t appear that any non-toxin homologous sequences were included in these datasets? I would think this would be necessary to determine the evolutionary selection pressures that toxins are experiencing by comparisons to non-toxic homologs.

2.     I am the most familiar with PAML, but this analysis is to be completed on ortholog sequences, and your PACAP dataset only includes one species – Desmodus rotundus, if this is true additional homologous sequences must be added to this analysis. This should also be true for the other selection analyses.

3.     It would be nice to see an introduction to these toxin families at the beginning of the manuscript to provide readers insight into these proteins and their functions. Details regarding what is already known about selection pressures on these toxins could be moved from the introduction into the discussion to be directly compared to the results from this study.

4.     A figure comparing the output of different selection tests would be interesting, shown for example as an alignment for a single protein family with amino acid residues highlighted that were identified as under positive selection for each program, colored in a way that would distinguish shared sites and sites only identified by one program. It is currently difficult to visualize this with just the amino acid residue numbers listed in tables.

5.     Are the sites implicated in the fitness of the animal near the active sites of toxins or exposed residues? Having a figure showing the toxin structure with identified amino acid residues under positive selection would also emphasize if mutations were within regions that would alter toxin activity.

Minor comments:

Line 15-17: These two sentences could be combined.

Line 180: “five branches”

Author Response

Dear editor

It is also clear from the reviewers’ comments that they would have liked to have more data, more information, more insights, into various aspects of our work – we agree with many of these, but in several instances we either simply do not have the data (and we are certain these data are not, or not publicly available) or if we were to address it in too much detail it would amount to speculation. 

We have updated the figure, we have updated some of the references, and we have reorganized some of the sections (moving them to the discussion for instance or into the methods).

Overall we feel that the manuscript has improved greatly and we hope you find it acceptable for publication.

Below are the detailed answers to all the questions, queries, comments and suggestions made by the reviewers.

Reviewer 1

Reviewer comment

Response

I feel that with some additional considerations, such as including a larger set of homologous sequences in these analyses and providing figures/visualizations of program outputs, this manuscript would be improved.

We have not included any additional homologous sequences (see below explanation) but we have now added additional figures (including a highlighted amino acid alignment on line 219-224 (figure 2) and  3D proteins on line 234-239 (figure 3))  to better illustrate our findings

It doesn’t appear that any non-toxin homologous sequences were included in these datasets? I would think this would be necessary to determine the evolutionary selection pressures that toxins are experiencing by comparisons to non-toxic homologs.

We follow the methodology of Sunagar and Moran (2015) “The Rise and Fall of an Evolutionary Innovation: Contrasting Strategies of Venom Evolution in Ancient and Young Animals” ensuring our findings can be compared and contrasted with theirs. Follow this, there is no need to include non-toxic homologs to determine and calculate selection pressures. We now have made this more clear in our Methods on line 578-579 by mentioning explicitly that we are following the methods of Sunagar and Moran.

I am the most familiar with PAML, but this analysis is to be completed on ortholog sequences, and your PACAP dataset only includes one species – Desmodus rotundus, if this is true additional homologous sequences must be added to this analysis. This should also be true for the other selection analyses.

Sungar and Moran’s method does not use non-toxic homologous and in some examples uses only one or two species representation for a toxin. In addition, it also uses only a few sequences in some examples as well. The limitations of the available data (as of October 2022) prevent us for bringing in additional mammal toxins as well. As indicated above we have provide more context and explanation in the introduction (lines 103-108) and the methods (lines 578-583).

 It would be nice to see an introduction to these toxin families at the beginning of the manuscript to provide readers insight into these proteins and their functions. Details regarding what is already known about selection pressures on these toxins could be moved from the introduction into the discussion to be directly compared to the results from this study.

We have followed this suggestion and moved parts of the Introduction to the Discussion (now lines 355-373) and we have added information on the various toxin families in the Introduction (lines 109-140)

A figure comparing the output of different selection tests would be interesting, shown for example as an alignment for a single protein family with amino acid residues highlighted that were identified as under positive selection for each program, colored in a way that would distinguish shared sites and sites only identified by one program. It is currently difficult to visualize this with just the amino acid residue numbers listed in tables.

Thank you for this suggestion, we have included an example of an alignment highlighted in the main text (line 219-224) and an additional ten figures of amino acid alignments highlighted are included in the supplementary (Supplementary 1, Figure S1-10)

Are the sites implicated in the fitness of the animal near the active sites of toxins or exposed residues? Having a figure showing the toxin structure with identified amino acid residues under positive selection would also emphasize if mutations were within regions that would alter toxin activity.

We agree that this is certainly worth further investigation, however we feel that this falls outside the scope of our current paper.

Line 15-17: These two sentences could be combined.

Thank you for pointing this out, we have updated accordingly (line 15-19)

Line 180: “five branches”

Thank you for pointing this out, we have updated accordingly by changing five to Five (line 249)

The manuscript “The fast and the furriest: investigating the rate of selection on mammalian toxins” is well-written and examines selection pressures across mammalian toxins with a level of detail that has yet to be done, making this a novel study. It is also quite interesting to see the output of multiple selection tests on the same dataset, although some variation is expected with such a small sample size.

We would like to thank Reviewer 1 for their encouraging comments, the time they have given to review this paper. Their feedback has been very useful and constructive.

Reviewer 2 Report

The manuscript is well written, and data appropriately presented. It brings an issue very interesting, and few explored in the toxinological field.

Below some suggestions to make the manuscript clearer:

In the data collection, it was performed a general search using the term ‘mammals’ to retrieve genes and proteins? Was there any specification about specie in the search? How were the criteria for species inclusion, as there are venomous mammalians not included in the results?

Moreover, non-toxic molecules were included in the study, but authors discussed all the results using the term ‘toxins’. Plasminogen activator, kunitz domains and others are not necessarily toxins, which does not invalidate the data and the conclusion, but its physiological importance, other than toxins, could be mentioned.

Although the supplementary table 1 presents data regarding toxins and species, it would be interesting to comment in the results how many species/classes were retrieved in the search, considered, and further analyzed, to the reader have an idea of the sample.

In the last sentence of the discussion, authors commented about comparison with different animal venoms. Considering that proteins like Kunitz-type inhibitors are present in several animal venoms/secretions, in all phylogenetic level, the amino acids alignment of proteins/genes used in this work and proteins from other species (such as cnidaria, insects, snakes and so on), would enrich the evolutionary discussion.

Author Response

Dear editor

It is also clear from the reviewers’ comments that they would have liked to have more data, more information, more insights, into various aspects of our work – we agree with many of these, but in several instances we either simply do not have the data (and we are certain these data are not, or not publicly available) or if we were to address it in too much detail it would amount to speculation. 

We have updated the figure, we have updated some of the references, and we have reorganized some of the sections (moving them to the discussion for instance or into the methods).

Overall we feel that the manuscript has improved greatly and we hope you find it acceptable for publication.

Below are the detailed answers to all the questions, queries, comments and suggestions made by the reviewers.

Reviewer 2

Reviewer comment

Response

In the data collection, it was performed a general search using the term ‘mammals’ to retrieve genes and proteins? Was there any specification about specie in the search? How were the criteria for species inclusion, as there are venomous mammalians not included in the results?

We have now included more information and more detail on how we conducted our search, including keywords and our search strategy, which is included in the methods (line 578-596) and results  (line 175-180).

Was there any specification about specie in the search? 

Thank you for pointing this out, we have now included this in the Methods (see above for specific lines, also relevant to making it clearer we are following the Sunagar and Moran, 2015 methodology)

How were the criteria for species inclusion, as there are venomous mammalians not included in the results?

Yes, please see above for specific lines

Moreover, non-toxic molecules were included in the study, but authors discussed all the results using the term ‘toxins’. Plasminogen activator, kunitz domains and others are not necessarily toxins, which does not invalidate the data and the conclusion, but its physiological importance, other than toxins, could be mentioned.

There is an ongoing debate what comprises a toxin, but we here have followed both Wong, et al., 2010 “Novel venom gene discovery in the platypus”, Low, et al., 2013 “ Dracula's children: Molecular evolution of vampire bat venom” and Kakumanu, et al., 2019 “Vampire Venom: Vasodilatory Mechanisms of Vampire Bat (Desmodus rotundus) Blood Feeding” we consider plasminogen activators and kunitz as toxins. We now have added a sentence in the methodology (line 584-588) where we explain that there is indeed a debate about what comprises a toxin and that this is an area that requires further work.  

Although the supplementary table 1 presents data regarding toxins and species, it would be interesting to comment in the results how many species/classes were retrieved in the search, considered, and further analyzed, to the reader have an idea of the sample.

We thank the reviewer and we have added information about the amount of species recovered and actually used to our Results (line 176 – 182)

In the last sentence of the discussion, authors commented about comparison with different animal venoms. Considering that proteins like Kunitz-type inhibitors are present in several animal venoms/secretions, in all phylogenetic level, the amino acids alignment of proteins/genes used in this work and proteins from other species (such as cnidaria, insects, snakes and so on), would enrich the evolutionary discussion.

We agree, but this would require a substantial amount of additional research and analysis and as indicated in the title, we here focus exclusively on mammals. 

The manuscript is well written, and data appropriately presented. It brings an issue very interesting, and few explored in the toxinological field.

We would like to thank Reviewer 2 for their encouraging comments, the time they have given to review this paper. Their feedback has been very useful and constructive.